# Model-based Cleaning of the QUILT-1M Pathology Dataset for Text-Conditional Image Synthesis

**Marc Aubreville**[1]                                    MARC.AUBREVILLE@THI.DE
**Jonathan Ganz**[1]                                      JONATHAN.GANZ@THI.DE
**Jonas Ammeling**[1]                                     JONAS.AMMELING@THI.DE
[1] *Technische Hochschule Ingolstadt, Ingolstadt, Germany*
**Christopher C. Kaltenecker**[2]         CHRISTOPHER.KALTENECKER@MUV.AC.AT
[2] *Medical University of Vienna, Austria*
**Christof A. Bertram**[3]                  CHRISTOF.BERTRAM@VETMEDUNI.AC.AT
[3] *University of Veterinary Medicine, Vienna, Austria*

## Abstract

The QUILT-1M dataset is the first openly available dataset containing images harvested from various online sources. While it provides a huge data variety, the image quality and composition is highly heterogeneous, impacting its utility for text-conditional image synthesis. We propose an automatic pipeline that provides predictions of the most common impurities within the images, e.g., visibility of narrators, desktop environment and pathology software, or text within the image. Additionally, we propose to use semantic alignment filtering of the image-text pairs. Our findings demonstrate that by rigorously filtering the dataset, there is a substantial enhancement of image fidelity in text-to-image tasks.

**Keywords:** histopathology, foundation model, data cleaning

## 1. Introduction

Foundation models are trained on vast amounts of data to build a latent space which reduces the amount of necessary labeled samples in a downstream task. The well-organized latent space enables learning of inherent relations within large and diverse input data. While initial approaches built foundation models on general-purpose images (Kirillov et al., 2023), recent adaptations focused on medical images (Ma et al., 2024) and specifically, histopathology images (Ikezogwo et al., 2024; Lu et al., 2024). However, many histopathology images have significant file sizes, with whole slide images often reaching several gigabytes per image. Additionally, anonymity concerns might hinder making pathology archives publicly available (Ganz et al., 2024). To circumvent the issue of data availability for the training of large-scale multimodal models, Ikezogwo *et al.* have proposed to use scraping from freely available online sources, such as Twitter communities, medical images available in articles retrieved from PubMed Central, and images and text automatically extracted from videos sourced from YouTube (Ikezogwo et al., 2024). This effort lead to a publicly available archive, denoted QUILT-1M, containing 653,209 images, linked to 1,017,708 captions. While this is undoubtedly a great source of data variability, the quality dimension of such a dataset can not be guaranteed by visual expert review at this scale.

Text-conditional image synthesis (a.k.a. text-to-image) has been vastly improved regarding image quality, mainly due to architectural advancements (Rombach et al., 2022)

and availability of large-scale training data, such as the 5 billion images of the LAION-5B dataset (Schuhmann et al., 2022). In the field of histopathology, these text-to-image models have multiple applications, e.g., diagnosing diseases from textual descriptions originating from pathologists, generating synthetic images for educational purposes, and even assisting in the development of new medical imaging techniques through simulated data generation. Especially for text-to-image models, the informativeness of data (Chen et al., 2024), including the quality and density of textual descriptions and the purity of images is of high importance. Hence, the quality of the model depends on how accurately the whole image with all its contents is described by the text. However, large-scale data, such as the QUILT-1M dataset, were created from educational resources, which often contain additional image parts (e.g., a narrator, pointers, highlights) that can be considered impurities in a text-conditional image synthesis setting. For this reason, we propose an automatic deep learning pipeline to filter out images with impurities.

| Impurity | Images affected (N=6,532) | Evaluation on test set (N=980) | | | |
|---|---|---|---|---|---|
| | | Accuracy | Recall | Specificity | ROC AUC |
| Narrator/person | 19.76% | 98.47% | 96.59% | 98.97% | 0.9962 |
| Desktop/window decorations/slide viewer | 35.96% | 95.76% | 98.88% | 97.76% | 0.9897 |
| Text/logo | 26.36% | 85.77% | 96.07% | 93.27% | 0.9683 |
| Arrow/annotations | 11.14% | 63.48% | 96.18% | 92.35% | 0.9108 |
| Image of insufficient quality | 6.34% | 72.55% | 98.06% | 96.73% | 0.9489 |
| Additional slide overview | 14.13% | 88.19% | 98.83% | 97.45% | 0.9840 |
| Additional buttons/control elements | 31.89% | 89.07% | 95.81% | 93.67% | 0.9733 |
| Multi-panel image | 7.88% | 67.50% | 99.33% | 96.73% | 0.9717 |
| Any of the above | 78.26% | 92.71% | 97.17% | 93.16% | 0.9481 |

Table 1: Image impurities annotated on a random 1% sample of QUILT-1M and detection performance on the hold out set by a ResNet50-D-based multi label impurity classifier.

## 2. Materials and Methods

**Annotation** To evaluate this for the QUILT-1M dataset, we manually evaluated 1% (6,532) of the images[1] of the dataset for common image impurities, see Table 1. The analysis reveals that only a minority (21.74%) of images are free from common additional image elements, such as a narrator in the image, additional text or logos on top of the image, multi-panel images, or graphical elements of the slide viewer software. To provide additional support for high quality image crops, we randomly sampled 2 times within the tumor from each of the images of the TCGA-BRCA cohort, yielding 2,072 impurity-free images.

**Model training** We split this sample using a 70%–15%–15% train-val-test split and trained a ResNet50-D-based multi-label impurity classifier using standard image augmentations and a cross-entropy loss. We performed early stopping and model selection based

---

1. All data and code available at: https://github.com/DeepMicroscopy/QuiltCleaner

| Dataset variant | generations for: `breast adenocarcinoma at 10x` | $FID_{MIDOG++}$ | $FID_{PLISM}$ |
|---|---|---|---|
| Original QUILT-1M |  | 247.24 
 (worst) | 175.74 
 (worst) |
| Impurity Classifier Filtered |  | 214.08 | 142.46 |
| Impurity Classifier +CONCH Score Filtered |  | 147.36 
 (best) | 111.38 
 (best) |

Figure 1: Example generations by the model trained on the dataset variants and the FID metric evaluated on 10,000 image crops retrieved from two datasets.

on the validation loss, and finally evaluated accuracy, specificity, recall and area under the ROC curve on the test set.

**Semantic refinement** We additionally used the CLIP score calculated by the CONCH model (Lu et al., 2024) to filter out the semantically better aligned half of the dataset (i.e., all image-text pairs exceeding the median CLIP score).

**Downstream task** We then filtered the remaining 99% of the dataset using the impurity classifier trained in the previous step and performed text-conditional image synthesis using a latent diffusion model (Rombach et al., 2022) pipeline, fine-tuning from the Stable Diffusion 2.1 weights for 15,000 iterations.

**Evaluation** We evaluated the model on all dataset variants using the conditional Fréchet Inception Distance (FID) on 10,000 generated images. For comparison, we used the MI-DOG++ dataset (Aubreville et al., 2023) and the PLISM dataset (Ochi et al., 2024), from which we derived the tumor-affected organ and the tumor type as prompts. We sampled randomly from PLISM and took random crops from the larger MIDOG++ images.

## 3. Results and Discussion

As shown in Table 1, our classifier was well capable of filtering the dataset, reaching an accuracy of 92.71% on impurity detection. In tasks involving text-to-image generation through diffusion-based models, we observed that models trained on the unfiltered dataset exhibited notable artifacts (see Fig. 1), which is also reflected by the higher FID scores. It is important to note that while FID is commonly used to assess image fidelity, its utility may be limited as the Inception network used to compute the score was trained on ImageNet rather than on a pathology-specific task. Our findings underscore the importance of filtering large-scale datasets sourced from public repositories prior to their utilization in text-to-image tasks.

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
