# OpenReview forum: "Model-based Cleaning of the QUILT-1M Pathology Dataset for Text-Conditional Image Synthesis"
_MIDL.io/2024/Short_Papers — MIDL 2024 Short Papers_

### Official Review · Reviewer_PKFX · 2024-04-25

**Confidence:** 4
**Final Rating:** 3.5

**Review:**

The paper focuses on the development of an automatic pipeline for filtering impurities in the QUILT-1M dataset, highlights the importance of data cleaning for text-conditional image synthesis tasks and proposes a model-based approach to filter out impurities.

Merits:
1. The paper addresses the critical issue of dataset quality and composition, providing a solution to filter out impurities in large-scale datasets, which is essential for improving the accuracy and fidelity of text-to-image generation models.
2. The proposed automatic pipeline utilizes a multi-label impurity classifier and semantic alignment filtering to effectively identify and filter out common impurities within the images, such as narrators, additional text or logos, and graphical elements.
3. The findings of the study demonstrate the significant improvement in image fidelity in text-to-image tasks after rigorously filtering the dataset, highlighting the practical relevance and impact of the proposed approach.

Limitations:
1. The study acknowledges the limitation of using the FID score, which was trained on ImageNet, to assess image fidelity in a pathology-specific task. This suggests the need for further research and development of pathology-specific evaluation metrics for text-to-image tasks.
2. The paper does not explicitly address the potential biases or ethical considerations associated with scraping images from freely available online sources, which could impact the generalizability and ethical implications of the dataset.

---

### Decision · Program_Chairs · 2024-04-26

Accept